# Single Cell RNA-Seq Identifies Immune-Related Prognostic Model and Key Signature-SPP1 in Pancreatic Ductal Adenocarcinoma

**DOI:** 10.3390/genes13101760

**Published:** 2022-09-29

**Authors:** Kai Chen, Qi Wang, Xinxin Liu, Feng Wang, Yongsu Ma, Shupeng Zhang, Zhijiang Shao, Yinmo Yang, Xiaodong Tian

**Affiliations:** 1Department of General Surgery, Peking University First Hospital, Beijing 100034, China; 2Department of Endoscopy Center, Peking University First Hospital, Beijing 100034, China; 3Department of General Surgery, Tianjin Fifth Centre Hospital, Tianjin 300450, China

**Keywords:** ScRNA-seq, prognostic model, SPP1, immune related signatures, pancreatic cancer

## Abstract

There are no reliable biomarkers for early diagnosis or prognosis evaluation in pancreatic ductal adenocarcinoma (PDAC). Multiple scRNA-seq datasets for PDAC were retrieved from online databases and combined with scRNA-seq results from our previous study. The malignant ductal cells were identified through calculating copy number variation (CNV) scores. The robust markers of malignant ductal cells in PDAC were found. Five immune-related signatures, including SPP1, LINC00683, SNHG10, LINC00237, and CASC19, were used to develop a risk score formula to predict the overall survival of PDAC patients. We also constructed an easy-to-use nomogram, combining risk score, N stage, and margin status. The expression level of SPP1 was related to the prognosis and immune regulators. We found that SPP1 was mainly expressed in ductal cells and macrophages in PDAC. In conclusion, we constructed a promising prognostic model based on immune-related signatures for PDAC using scRNA-seq and TCGA_PAAD datasets.

## 1. Introduction

Pancreatic ductal adenocarcinoma (PDAC) is a challenging disease, and the prognosis of patients has not been improved for decades, with a five-year overall survival (OS) rate of 9–10% [1]. The dismal prognosis of PDAC is mainly ascribed to a low surgical resection rate, chemotherapy resistance, and a lack of reliable biomarkers for early diagnosis. Radical resection and perioperative chemotherapy are the best strategies for curing PDAC. However, around 80% of patients lose their chance of surgery due to vascular invasion and/or distant metastasis at the time of diagnosis [2,3]. Even for those following radical resection, over half of patients have tumor recurrence or distant metastasis within two years, with a five-year OS rate of only 25–30% [4]. More aggressive adjuvant treatment might improve prognosis and delay the recurrent time, especially for selected patients with worse biological behavior. Thus, the accurate prognostic prediction of PDAC is essential for individual treatment.

Currently, immunotherapy has become a novel and efficient clinical strategy for treating a variety of malignancies. However, the overall treatment effect of immunotherapy is not ideal for patients with PDAC, and the immunotherapy response varies a lot [5,6]. Therefore, it is imperative to elucidate the immune microenvironment heterogeneity to pave the way to personalized immunotherapy for PDAC. Recently, single cell RNA-seq (scRNA-seq) revealed tumor microenvironment heterogeneity in an unprecedented resolution [7]. The scRNA-seq is a technology used to acquire the transcriptomic information of each cell, consisting of single cell suspension, library preparation, sequencing, and data analysis. With sequencing technology and bioinformatics, scRNA-seq makes it possible to delineate the cell landscape, identify new cell types, develop the trajectory, and perform the key immune-related events in the tumor microenvironment [8]. Peng et al. [9] performed scRNA-seq for 24 PDAC and 11 normal pancreatic specimens and found two distinct ductal subpopulations with different malignancies by calculating the copy number variation (CNV) scores. Elyada et al. [10] identified new cancer-associated fibroblast (CAFs) subtypes by conducting human and mouse CAF scRNA-seq, which is helpful for developing new targeted treatment for the specific CAF subtypes. Lin et al. [11] described the compositions of primary and malignant PDAC tissue using scRNA-seq. In addition, the scRNA-seq and spatial transcriptomics were combined to explore the characteristics of CAFs [12]. However, transcriptional profiles of immune and tumor cells in PDAC remain to be explored using scRNA-seq analysis.

Multiple scRNA-seq and bulk-seq datasets are available online and remain to be further investigated. Various diagnostic and prognostic models for PDAC based on gene expression and mutation have been proposed using bulk-seq datasets [13,14,15,16]. Nevertheless, the construction of a prognostic model by integrating multiple scRNA-seq and bulk-seq datasets has not been reported. Recently, immune-related genes and LncRNAs for predicting the survival and immunotherapy response were under the spotlight and had good performance in other tumors, whereas it is neglected in PDAC [17]. Compared with bulk-seq, scRNA-seq could provide more accurate signatures to construct a prognostic model. However, the bulk-seq dataset had a larger sample size and complete follow-up data. Therefore, it is necessary to construct a prognostic model for PDAC by integrating scRNA-seq and bulk-seq datasets.

In this study, we aimed to establish a prognostic model based on immune-related signatures by integrating scRNA-seq and TCGA_PAAD datasets. The malignant ductal cells were identified using single-cell CNV analysis. The robust marker genes of malignant ductal cells in PDAC were identified using common differential expression genes (DEGs) from multiple human scRNA-seq datasets. Next, we selected immune-related DEGs and LncRNAs by univariate cox regression analysis and LASSO-penalized cox regression analysis. Multivariate cox regression analysis was conducted to develop and validate a prognostic model for PDAC. In addition, we examined the expression and distribution of the key prognostic signature SPP1 in PDAC through analyzing our previous scRNA-seq data and multicolor immunohistochemistry (IHC).

## 2. Materials and Methods

### 2.1. Patient Cohorts and Study Design

The scRNA-seq datasets were retrieved from GSA (Genome Sequence Archive; CRA001160), CNGBdb (China National GeneBank DataBase; CNP0001768), GEO (Gene Expression Omnibus; GSE129455, GSE125588, GSE154778), and our previous study. RNA-seq and clinical data of TCGA_PAAD were downloaded from the TCGA database updated to 20 April 2021. A total of 36 postoperative PDAC and matched adjacent normal pancreatic specimens were retrieved from the Department of General Surgery of Peking University First Hospital for verifying the expression and distribution of SPP1. This study was approved by the Ethics Committee of Peking University First Hospital (Approval No. 2019–147) and was conducted in accordance with the Declaration of Helsinki. Written informed consent was obtained from all participants.

This study employed a three-phase design. (a) The malignant ductal cells in PDAC were identified by inferCNV analysis using the scRNA-seq datasets. We compared malignant and normal ductal cells and identified common immune-related DEGs. (b) The prognostic model was constructed using immune-related genes and LncRNAs by univariate and multivariate cox regression analyses using the TCGA_PAAD dataset. An easy-to-use nomogram to predict the OS of PDAC patients was constructed as well. (c) We examined the expression and distribution of SPP1 among cell subpopulations in PDAC by scRNA-seq and multiple color IHC.

### 2.2. Single Cell RNA-Seq Analysis

Raw data (fastq files) were downloaded from the open databases or generated with BCL files from Illumina Novaseq6000 platform using Illumina-implemented software bcl2fastq (v2.19.0.316). cDNA reads were aligned to human or mouse reference genomes (GRCh38 or mm10), according to the previous study [18]. Low-quality cell and gene filtering and bar-code and UMI counting were performed using the CellRanger software (v6.1.2) to obtain the filtered gene-cell matrixes. Then, gene-cell matrixes were imported into R software to further filter out low-quality cells (<500 genes/cell, >20% mitochondria genes, <1000 transcripts/cell) and genes (<10 cells/gene) using the Seurat R package (v3.2.3). Gene expression levels were normalized (LogNormalize) with the NormalizedData function. A total of 2000 highly variable genes were selected and used to perform the PCA reduction dimension. The t-distributed stochastic neighbor embedding (t-SNE) was performed. Doublets were identified using the DoubletFinder R package (v2.0.3), assuming that it was around a 5% doublet formation rate to the loaded cells per specimen in a droplet channel. The Soupx R package was applied to reduce the ambient mRNA contamination. In addition, the Harmony R package (https://github.com/immunogenomics/harmony, accessed on 28 March 2022) was used to integrate gene-cell matrixes derived from distinct specimens. Cell type was identified by matching the cluster-specific genes with known signatures of cell populations reported in previous studies and the CellMarker database.

### 2.3. Cell Culture

As described in previous studies [19,20], the human ductal cell line (hTERT-HPNE) and pancreatic cancer cell lines, MIA PaCa-2, AsPC-1, PANC-1, T3M4, and BxPC-3 were bought from ATCC. Pancreatic cancer line PaTu8988 was provided from PharmLab (PharmLab, Beijing, China). MIA PaCa-2, PANC-1 (DMEM, Gibco, Cat. no. C11995500), AsPC-1, BxPC-3, and T3M4 (RPMI 1640, Gibco, Cat. no. C11875500) were cultured in cell culture dishes (NEST Biotechnology, Wuxi, China) in a humidified incubator at 37 °C with 5% CO_2_. The hTERT-HPNE was cultured in 75% DMEM without glucose and 25% M3 Base (Incell, Cat. no. M300F-500) supplemented with 10 ng/mL human recombinant EGF (CST, Cat. no. 72528), 5.5 mM D-glucose (1 g/L), 5% FBS, and 0.75 mg/mL puromycin (MCE, Cat. no. 58-58-2). All cell lines were authentic by short tandem repeats profile. Cells were maintained in a cell incubator (ThermoFisher, Waltham, MA, USA) with 5% CO_2_ and 20% O_2_ for a normoxic condition and incubator chamber (Billups-rothenberg, San Diego, CA, USA) flushed with 5% CO_2_ and 95% N_2_ for hypoxic condition.

### 2.4. Development of Prognostic Model

Gene expression matrix and clinicopathologic data were imported into R software (v4.0.3). Univariable and multivariable cox regression analyses were performed to evaluate the clinical values of the selected DEGs using the ‘survival’ R package (v3.2.7). Risk score  =  h0 × e^∑i  =  0n exp (). Patients were classified into two groups (high vs. low risk) according to the median risk score. The KM and ROC curves were used to evaluate the value of the prognostic models using the ‘survivalROC’ R package (v1.0.3). Next, a nomogram was developed to predict the overall survival of patients with PDAC by combining the risk score and clinicopathologic features using the ‘rms’ R package (v6.2.0). The calibration curve was also drawn to evaluate the accuracy of nomogram-predicted patients’ survival.

### 2.5. Tumor Immune Infiltration Analysis

The immune score of each subject in TCGA_PAAD was calculated according to the ESTIMATE algorithm [21]. We compared the immune scores between the high- and low-risk groups and explored the effect of immune scores on the OS of PDAC patients. The CIBERSORT method was used to reveal the immune cell composition difference between the high- and low-risk groups [22]. In addition, the GEPIA tool was utilized to perform deconvolution-based analyses for each subject of TCGA_PAAD and GTEx_pancreas, which included three bioinformatics platforms: CIBERSORT, EPIC, and quanTIseq. The proportion of cell subpopulations in PDAC and normal pancreatic tissues was inferred [23].

### 2.6. Multiple Color IHC

To explore the expression and distribution of SPP1 in PDAC and normal pancreatic tissues, formalin-fixed paraffin-embedded (FFPE) sections from cancerous and adjacent normal tissues were subjected to multiple IHCs using PANO Multiplex IHC kit (Panovue, Beijing, China, cat# 0079100100). Briefly, sections were deparaffinized in fresh xylene, rehydrated in a decreasing concentration of ethanol, then washed three times with PBS. The antigen retrieval was performed with the microwave heating method and cooled down for at least 10–15 min in an ice-water bath. Blocking was performed with a blocking solution (Panovue, China, cat# 0018001120) for 10 min at room temperature, followed by incubation with a primary antibody for 30 min, a secondary antibody for 10 min, and TSA Opal fluorophores for 10 min. Antigen retrieval, blocking, primary and antibody incubation, TSA Opal fluorophore staining were repeated for each marker. Finally, all sections were counterstained with DAPI (Sigma-Aldrich, USA, cat# D9542) for 5 min and mounted. Sections were scanned using the panoVIEW VS200 (Panovue, Beijing, China). Images captured and quantitative analysis were conducted using the HALO software. The following primary antibodies were used: SPP1 (Abcam, Cambridge, UK, cat# ab214050), CD68 (ZSGB-Bio, Beijing, China, cat# ZM0060), and PanCK (CST, Boston, MA, USA, cat# 4545).

### 2.7. Statistical Analysis

All statistical analyses were performed in the R tool (v.4.0.3). RT-qPCR assays were performed in three replicates and repeated three times independently. The KM method and the corresponding log-rank test were performed to identify the prognostic value of marker genes. The calibration curve was performed to evaluate the accuracy of the model. For continuous variables, the independent-sample t-test and Mann-Whitney *U* test was performed to compare the means between two groups. Correlation analysis was conducted using the Pearson/Spearman test. Statistical significance was defined as * *p* < 0.05, ** *p* < 0.01, and *** *p* < 0.001.

## 3. Results

### 3.1. ScRNA-Seq Identified Immune Related DEGs for PDAC

To identify transcriptional profile differences between tumor cells and normal ductal cells in PDAC, we conducted scRNA-seq for cancerous and adjacent tissues from patients with PDAC in our institution and downloaded scRNA-seq datasets from the Genome Sequence Archive (GSA) and China’s National GeneBank DataBase (CNGBdb) databases. The t-distributed stochastic neighbor embedding (t-SNE) was used to delineate the cell atlas for different datasets (Appendix A). Most of cell subpopulations in PDAC and normal pancreatic tissues, including T cells, fibroblasts, macrophages, ductal cells, endothelial cells, stellate cells, acinar cells, B cells, plasma, mast cells, neutrophils, Schwann cells, and endocrine cells, were identified by scRNA-seq analysis. Our previous study isolated ductal cell subpopulations, inferred somatic large-scale chromosomal CNVs, and calculated CNV scores according to the reference normal cells, such as endothelial cells, macrophages, and T cells. Ductal cell subpopulations with high CNV scores were defined as malignant ductal cell subpopulations; therefore, they were marked as tumor 1/2/3/4/5 (Appendix A). We compared normal and malignant ductal subpopulations, and identified many DEGs (Figure 1B–D). Furthermore, common DEGs (Tumor 1/2/3/4/5 vs. normal ductal subpopulations) were identified for each scRNA-seq dataset (Figure 1E–G). The Venn diagram showed the intersection of common DEGs among three datasets (Figure 1H). A total of 28 common DEGs were found, which represented the most significant differences between tumor cells and normal ductal cells in PDAC. Seven DEGs were related to immune regulation according to the Immunology Database and Analysis Portal (ImmPort) database. A correlation analysis was performed to select the most relevant LncRNAs using the TCGA_PAAD dataset. Finally, seven immune-related DEGs (IRDEGs) and 70 LncRNAs were included in developing the prognostic model.

### 3.2. The Construction and Internal Validation of the Prognostic Model for PDAC

The gene expression matrix and clinical follow-up data were downloaded and integrated using the TCGA_PAAD dataset. Univariate cox regression analysis was conducted to screen out prognosis-related signatures. A total of 30 OS-related IRDEGs and LncRNAs were found. To avoid the prognostic model overfitting, LASSO-penalized cox regression analysis was performed to further select 10 signatures for model construction (Figure 1I,J). Then, 177 subjects in the TCGA_PAAD dataset were randomly divided into a train set and validation set in 2:1. Multivariate cox regression analysis was used to construct a prognostic model in the train set (Appendix A). Five immune-related signatures, including SPP1, LINC00683, SNHG10, LINC00237, and CASC19, were used to develop a risk score formula to predict the overall survival of PDAC patients. The risk score of each subject was calculated as follows: risk score (t) = h0 (t) * exp (0.1775 * SPP1–0.3990 * LINC00683–0.3215 * SNHG10–0.3744 * LINC00237 + 0.1537 * CASC19).

Subjects were classified into a low-risk group or high-risk group according to the median risk score (Figure 2C). The Kaplan–Meier curve (KM) indicated that subjects with higher risk scores had significantly worse prognoses than those with lower risk scores (*p* = 2.877 × 10^−6^) (Figure 2A). The time-dependent receiver operating characteristic curve (ROC) was utilized to assess the validity of the risk score model to predict 1-year, 1.5-year, and 2-year Oss, and the area under curve (AUC) values were 0.709, 0.77, and 0.787, respectively (Figure 2B). The risk curve showed that the higher the risk score subjects had, the shorter OS they were inclined to have (Figure 2D). Next, the internal validation was performed. In line with results in the train set, subjects in the high-risk group had worse OS in the validation set (*p* = 1.361 × 10^−2^; AUC: 1-/1.5-/2-year OSs: 0.667/0.726/0.818) and all set (*p* = 1.361 × 10^−2^; AUC: 1-/1.5-/2-year OSs: 0.681/0.751/0.798) (Figure 2E–L).

To further test the performance of the established prognostic model in different age and N stage subgroups, subjects were divided into two subgroups (age > 65 and age ≤ 65; N0 and N1). Consistent with previous results, subjects in the high-risk group had worse OS than those in the low-risk group for subgroup analyses. For the age > 65 subgroup, the *p* value was 4.362 × 10^−3^ in the KM curve, and 1-/1.5-/2-year AUC values were 0.637/0.71/0.756 (Appendix A). For the age ≤ 65 subgroup, the *p* value was 8.972 × 10^−5^ in the KM curve, and 1-/1.5-/2-year AUC values were 0.718/0.791/0.818 (Appendix A). For the N0 subgroup, the *p* value was 3.49 × 10^−2^ in the KM curve, and 1-/1.5-/2-year AUC values were 0.639/0.764/0.803 (Appendix A). For the N1 subgroup, the *p* value was 2.318 × 10^−3^ in the KM curve, and 1-/1.5-/2-year AUC values were 0.663/0.669/0.718 (Appendix A).

### 3.3. Nomogram to Predict the Survival of Patients with PDAC

The clinicopathological characteristics and risk scores of each subject in TCGA_PAAD were combined. The univariate cox regression analysis indicated that risk score (HR = 2.16; 95% CI: 1.50–3.09, *p* = 0.0000), margin status (HR = 1.72; 95% CI: 1.02–2.92, *p* = 0.0431), and N stage (HR = 1.95; 95% CI: 1.03–3.67, *p* = 0.0398) were significantly associated with the OS (Figure 2M). However, the multivariate cox regression analysis showed that risk score was the only independent prognostic factor of PDAC (HR = 2.01; 95% CI: 1.4–2.9, *p* = 0.0002) (Figure 2N). In addition, an easy-to-use prognostic nomogram, including N stage, margin status, and risk score, was established (Figure 2O). The subject with higher total points was predicted to have shorter 2-year and 3-year OSs. There was a good correlation between nomogram-predicted OSs and actual OSs, showing the accuracy of the established nomogram (Figure 2N).

### 3.4. The Prognostic Model Was Associated with Tumor Immune Infiltration Status

The type of tumor immune infiltration was a good predictor for prognosis and immunotherapy response [26,27]. To explore the correlation between the established prognostic model and the tumor immune infiltration status in PDAC, the immune infiltration score of each subject was calculated. Subjects in the high-risk group had higher immune infiltration scores compared to the low-risk group (*p* = 0.003) (Figure 3A). The KM curve showed that subjects with higher immune infiltration scores had a significantly worse OS (1.241 × 10^−2^) (Figure 3B). PDAC tissues from the high-risk group had a higher proportion of Tregs (*p* < 0.05), which might contribute to the dismal prognosis (Figure 3C). In addition, multiple deconvolution-based analyses for the proportion of cell subpopulations in PDAC and normal pancreatic tissues showed that there was a significant difference between PDAC and normal pancreatic tissues in immune cell compositions, and the proportions of Tregs and M2_macrophage were significantly higher in PDAC than normal pancreatic tissues for CIBERSORT and quan TIseq, indicating that PDAC underwent the process of suppressive immune regulation (Appendix A). PDAC also had more cancer-associated fibroblasts (CAFs) for EPIC.

### 3.5. Pan-Cancer Landscape of SPP1 Was Delineated

SPP1, a key signature in the established prognostic model, plays a pivotal role in the tumor immune microenvironment and tumor progression [28,29,30]. SPP1 is also a promising biomarker for various tumors [31,32,33]. The pan-cancer landscape of SPP1 for diagnosis, prognosis, and tumor immune infiltration was investigated. SPP1 was significantly upregulated in various malignancies, such as breast cancer, cholangiocarcinoma, glioblastoma, kidney carcinoma, pancreatic cancer, brain lower grade glioma, etc. (Figure 3D). The KM curves showed that patients with a higher expression of SPP1 had worse OSs in many tumors, including adrenocortical carcinoma, cervical carcinoma, head and neck carcinoma, lower grade glioma, liver hepatocellular carcinoma, and pancreatic cancer (Figure 3E). Furthermore, the expression level of SPP1 was related to the abundance of tumor-infiltrating Tregs and macrophages (Figure 3F,G). There was a significant difference of expression of SPP1 between responders and non-responders in various checkpoint blockade therapy datasets (Figure 3H). The expression of SPP1 was also correlated to multiple immunoinhibitors (CSF1R, HAVCR2, IL10, PDCD1LG2, TGFB1) and immunostimulators (CD80, CD86, IL2RA, TNFSF4/9) (Appendix A).

### 3.6. ScRNA-Seq Identified the Expression of SPP1 in Various Cell Subpopulations of PDAC

Recently, scRNA-seq showed great advantages in uncovering transcriptional profiles of tissues in single cell resolution. Multiple human and mouse scRNA-seq datasets for PDAC were retrieved from online databases. In our previous study, we described tissue landscapes and identified cell subpopulation identifications. Here, the expression level of SPP1 among various cell subpopulations was shown by violin plot. We found that many cell subpopulations, especially macrophage and ductal cells, had an enriched expression of SPP1 in human and mouse PDAC tissues (Figure 4A,B). In addition, PDAC had the higher expression of SPP1 than normal pancreatic tissue (Figure 4C). However, no significant difference was found in the expression of SPP1 for PDAC patients with different stages (Figure 4D). Notably, M2_macrophage in PDAC had a higher expression of SPP1 than those in normal pancreatic tissue. In summary, SPP1 in M2_macrophage might be a better diagnostic and prognostic biomarker for PDAC.

To further explore the expression of SPP1 in distinct macrophage subpopulations, the Gene-Cell matrix of macrophages in our scRNA-seq dataset was isolated for t-SNE analysis. The eight original clusters were found and then integrated into four macrophage subpopulations: macrophage 1–4 (Figure 5A,B,D). There was a significant difference in gene expression profiles among the four macrophage subpopulations (Figure 5C). Macrophage 2 had enriched expression of many chemokines and ligands, including CCL4, CCL4L2, CCL3, and CCL3L1, which were involved in recruiting immune cells to promote tumor development [35,36]. Next, we tested the expression of markers of different macrophage types among macrophage 1–4 subpopulations (Figure 5E). The common macrophage marker AIF1 was expressed in all isolated macrophages, identifying their cell identification. The M2_macrophages and CD169+ macrophages were found based on the expression of IL4R and SIGLEC1. The SPP1+ macrophages were distributed among macrophage 1–4 subpopulations. There were few macrophages that expressed both IL4R and SPP1.

### 3.7. The Distribution of SPP1 in PDAC Was Verified by Multicolor IHC Stating

To further verify the distribution of SPP1 in PDAC, we tested the expression of SPP1 in PDAC cell lines. The RT-qPCR results showed that many PDAC cell lines, except for MIA PaCa-2, expressed the higher levels of SPP1 compared to HPNE (Figure 6A). Similar results were observed in western blot (Figure 6B). The upregulation of SPP1 was found in the MIA PaCa-2 and BxPC-3 under a hypoxic condition (Figure 6C). In addition, the paired cancerous and adjacent tissues of PDAC patients were examined. We found that cancerous tissues had significantly enriched the expression of SPP1 compared to the adjacent tissues (Figure 6D). The results of the multicolor IHC showed that SPP1 was expressed in various cell types in PDAC tissues, and PanCK+/SPP1+ and CD68+/SPP1+ cells were observed, indicating that SPP1 was enriched in ductal cells and macrophages.

## 4. Discussion

In this study, we constructed a promising immune-related prognostic model for PDAC by integrating scRNA-seq and TCGA_PAAD datasets. The robust marker genes of malignant ductal cells in PDAC were identified using multiple human scRNA-seq datasets. We also explored the correlation between the prognostic model and the tumor immune infiltration status. Through scRNA-seq and multicolor IHC analyses, the expression and distribution of the key prognostic signature SPP1 in PDAC were delineated. SPP1 was mainly expressed in macrophage and ductal cells.

Recently, scRNA-seq technology shows great advantages in revealing the transcriptional profiles of cell subpopulations in single-cell resolution. Through calculating the CNV score of each cell, we identified malignant ductal cells in PDAC tissues. Consistent with previous studies [9,18,37,38], there were significant differences among malignant ductal cell subpopulations from different patients, whereas normal ductal cell subpopulations were homogeneous, suggesting a patient-derived heterogeneity. A growing body of study explored the transcriptional profile differences between PDAC and normal pancreatic tissues based on bulk-seq and microarray [39,40,41,42]. However, the transcriptional profile of tumor cells in PDAC were covered by other main cell subpopulations based on the whole tissue gene expression profiling. In this study, we directly compared malignant and normal ductal cells by scRNA-seq. The common DEGs between many patient-derived malignant and normal ductal cell subpopulations were identified. Moreover, the DEG intersections from multiple scRNA-seq datasets were collected to obtain robust results.

It is necessary to estimate the long-term survival of patients with PDAC for individualized therapy. Recent studies developed various tumor molecular classifications to predict the survival outcomes of PDAC patients based on gene expression and mutation profiling, whereas LncRNAs were largely neglected [43,44,45]. In this study, we constructed a risk-score prognostic model according to immune-related genes and LncRNAs. Compared with clinicopathological characteristics, risk score had a higher HR value and was the independent prognostic factor in the multivariate cox regression analysis, which indicated the significantly prognostic value. We also established an easy-to-use nomogram to predict 2-year and 3-year OSs.

The SPP1 gene is located in 4q22.1 and encodes the secreted phosphoprotein 1, which is involved in cell adhesion to the extracellular matrix (ECM) [46,47], tumor proliferation, migration [48], chemoresistance [49], and macrophage polarization [38,39]. SPP1 was a promising biomarker for tumor diagnosis and prognosis [31,50,51]. However, the expression and distribution of SPP1 in the different cell subpopulations of PDAC tissues are poorly understood. In this study, we found that the expression of SPP1 was higher in multiple tumor tissues, and patients with higher SPP1 expression had a worse prognosis. SPP1 was related to the abundance of tumor-infiltrating Tregs and macrophages in many tumor tissues. Through exploring human and mouse scRNA-seq datasets for PDAC, we found that SPP1 was mainly enriched in ductal and macrophage subpopulations. Multicolor IHC stating also verified the distribution of SPP1 in PDAC tissue. Nevertheless, further studies are needed to explore the expression and function of SPP1 in distinct macrophage subpopulations. In addition, SPP1 was found to be overexpressed in many tumors, including GBM and ovarian cancer; therefore, the specificity of SPP1 as a biomarker of PDAC remains to be explored. The roles of SPP1 in mediating tumor progression also need to be revealed in the future.

Overall, we established a promising prognostic model for PDAC. However, this study has several limitations. First, external validation was not conducted. The online microarray datasets for PDAC lacked the LncRNAs included in our model. We will collect cancerous and adjacent tissues from PDAC patients for transcriptome sequencing to verify the validity of this model. Second, the prognostic model was constructed based on the TCGA_PAAD dataset, instead of scRNA-seq, because scRNA-seq cohorts for PDAC had a small sample size and lacked complete follow-up information. Large-scale scRNA-seq remains to be conducted. Third, both bulk-seq and scRNA-seq are dependent on postoperative surgical specimens. It is more important in clinical practice to develop a prognostic model based on preoperative biomarkers, including circulating tumor cells, exosomal miRNAs, and metabolites.

In conclusion, we constructed a promising prognostic model based on immune-related signatures for PDAC using scRNA-seq and TCGA_PAAD datasets. In addition, we examined the expression and distribution of SPP1 in PDAC.

## Figures and Tables

**Figure 1 genes-13-01760-f001:**
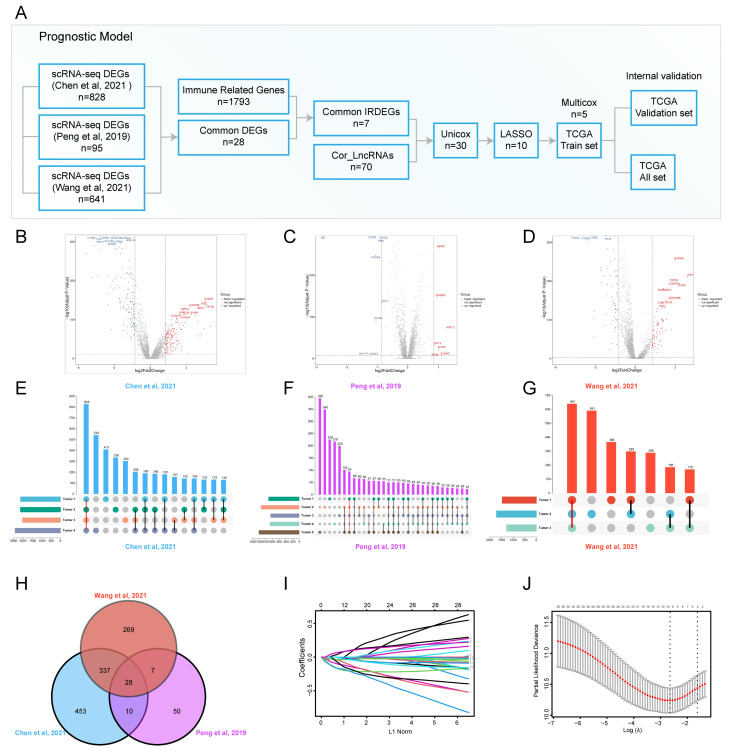
**scRNA-seq identified common DEGs for prognostic model construction [9,24,25].** (**A**) Graphical scheme of the construction of the prognostic model; (**B**–**D**) volcano plots showing the transcriptional profile difference between malignant and normal ductal cells; (**E**–**G**) common DEGs between malignant and normal ductal cell subpopulations are shown in an Upset plot; (**H**) Venn diagram plot showing DEG interaction among multiple scRNA-seq datasets; (**I**,**J**) variable selection using the LASSO regression, the correlation between coefficients and the number of variables (**I**), and the first dashed line showing the selected cutoff value, indicating minimal deviance (**J**).

**Figure 2 genes-13-01760-f002:**
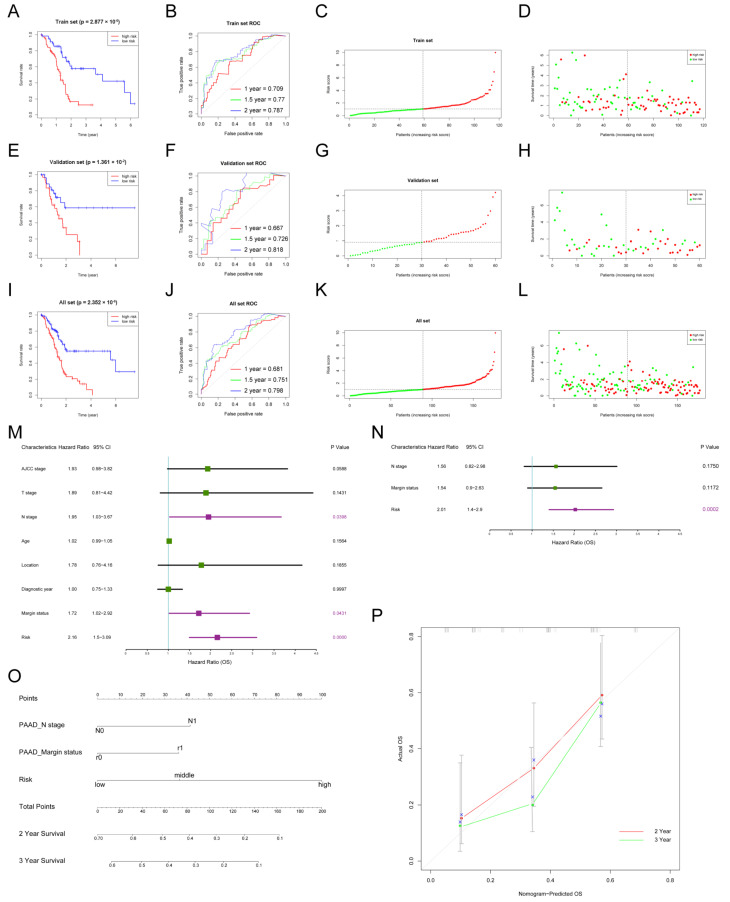
**The construction of the risk score model and nomogram to predict the OS of PDAC patients.** (**A**–**D**) Construction of the prognostic model in the train set in TCGA_PAAD: KM curve showing different OSs between high- and low-risk groups (**A**), ROC curve was used to evaluate the accuracy of the prognostic model for 1-/1.5-/2-year OSs (**B**), risk score distribution of subjects in the train set (**C**), and survival status scatter plot (**D**); (**E**–**H**) internal validation of the prognostic model in the validation set; (**I**–**L**) internal validation of the prognostic model in the all set; (**M**,**N**) univariate and multivariate cox regression analysis of risk factors of OS in PDAC; purple boxes represent *p* < 0.05 in the forest plot; (**O**) the nomogram was used to predict 2/3-year OSs of PDAC patients; (**P**) calibration curve was drawn to evaluate the consistency of predicted and actual OSs.

**Figure 3 genes-13-01760-f003:**
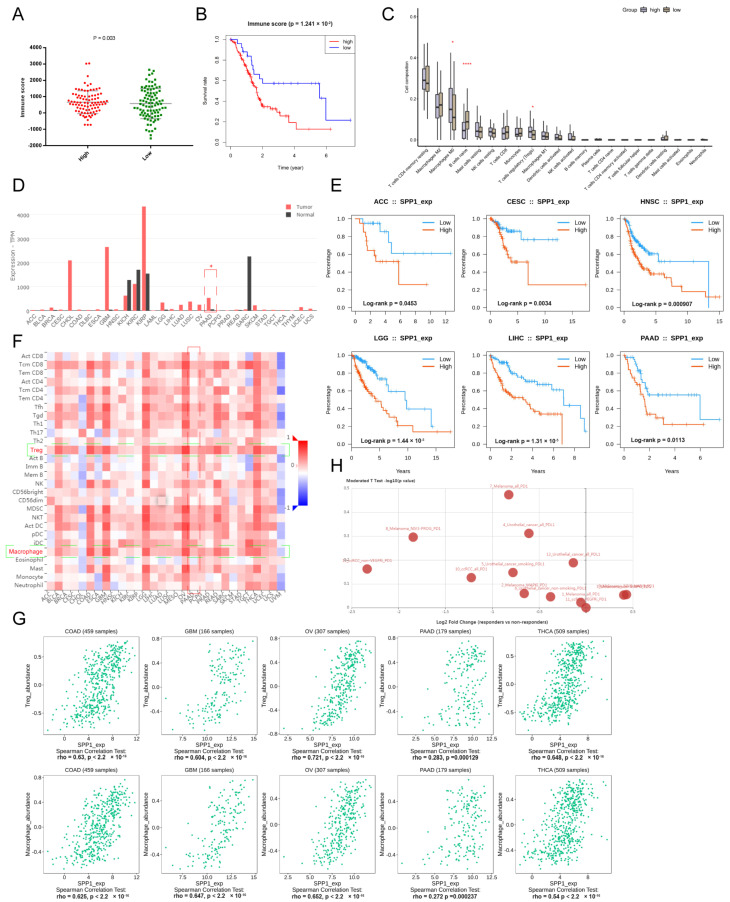
**The landscape of key signature–SPP1.** (**A**) The different immune scores between high- and low-risk groups; (**B**) KM curve showing the OS of patients between the high- and low-immune score groups; (**C**) the immune cell composition of high- and low-risk groups are shown in box plots. *, **** represent *p* < 0.05 and 0.0001; (**D**) the expression level of SPP1 among various tumors (red bars represent a tumor and black bars represent normal tissues); (**E**) KM curves showing the prognostic value of SPP1 among various tumors; (**F**) the heatmap showing the correlation of the expression of SPP1 and immune cell subpopulations, targeted cell types were labeled with green frame; (**G**) the scatter plots showing the relationship between the expression of SPP1 and Tregs and macrophages among various tumors; (**H**) the expression of SPP1 between responders and non-responders in various checkpoint blockade therapy datasets.

**Figure 4 genes-13-01760-f004:**
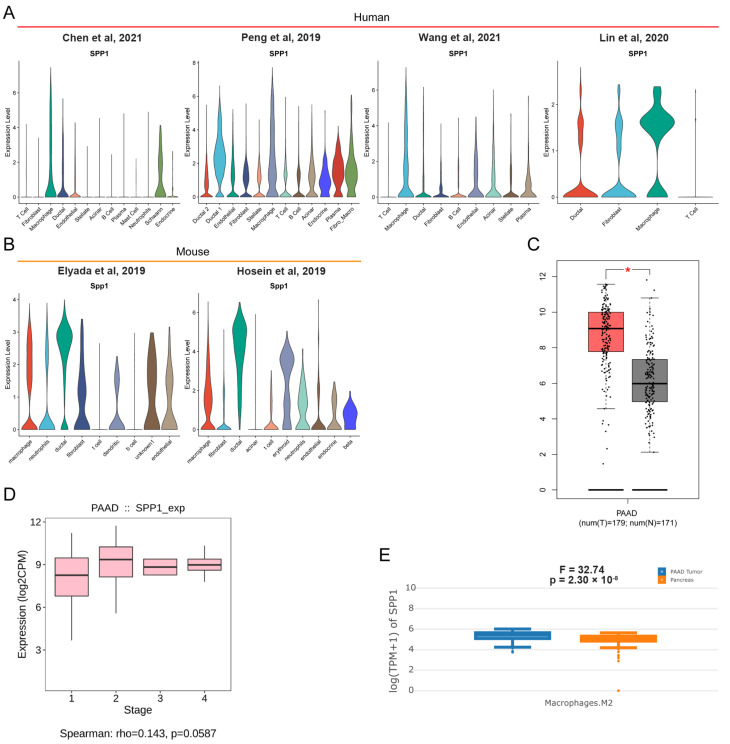
**scRNA-seq revealed the distribution of SPP1 in PDAC [9,10,11,24,25,34].** (**A**,**B**) Violin plots showing the expression level of SPP1 among distinct cell subpopulations in human and mouse PDAC scRNA-seq datasets; (**C**) the different expression levels of SPP1 in tumor and adjacent normal tissues in TCGA_PAAD and GTEx_pancreas. * represent *p* < 0.05; (**D**) the expression level of SPP1 among subjects with different stages in TCGA_PAAD; (**E**) M2_macrophage in PDAC had a higher expression of SPP1 than those in normal pancreatic tissues.

**Figure 5 genes-13-01760-f005:**
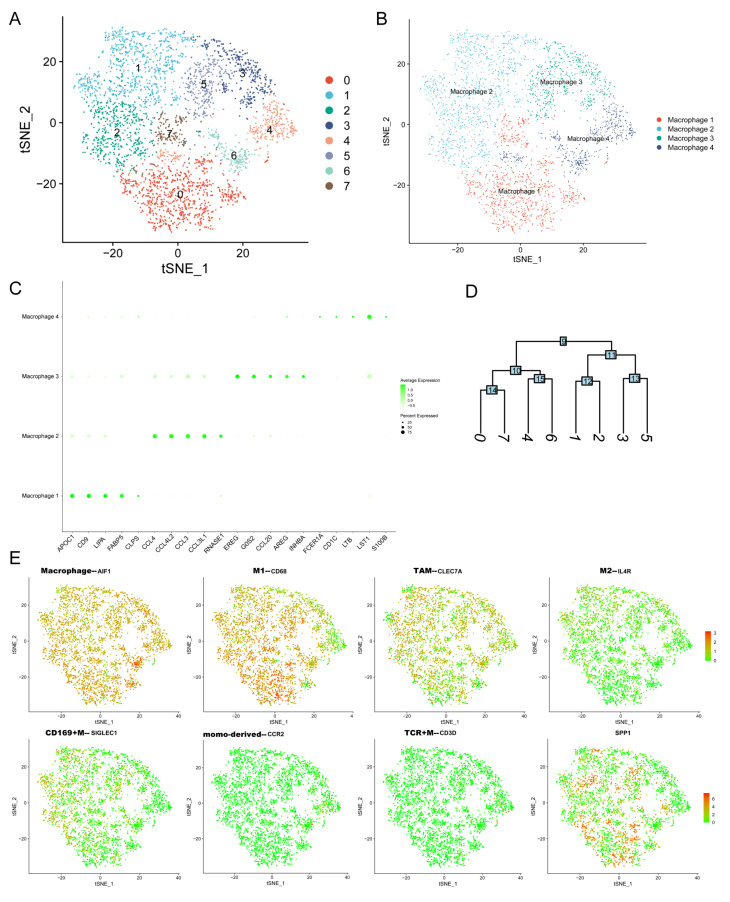
**The expression of SPP1 among distinct macrophage subpopulations.** (**A**,**B**) The t-SNE plots showing the original cluster (**A**) and named macrophage subpopulations; (**C**) dot plot showing the top 5 marker genes across macrophage subpopulations (size of dots represents the proportion of cells expressing a particular marker, and the intensity of color indicates the average expression level); (**D**) clustering diagram showing the similarity of clusters; (**E**) the expression levels of markers of many macrophage subtypes are shown in t-SNE plot (red and green dots represent high and low expression levels).

**Figure 6 genes-13-01760-f006:**
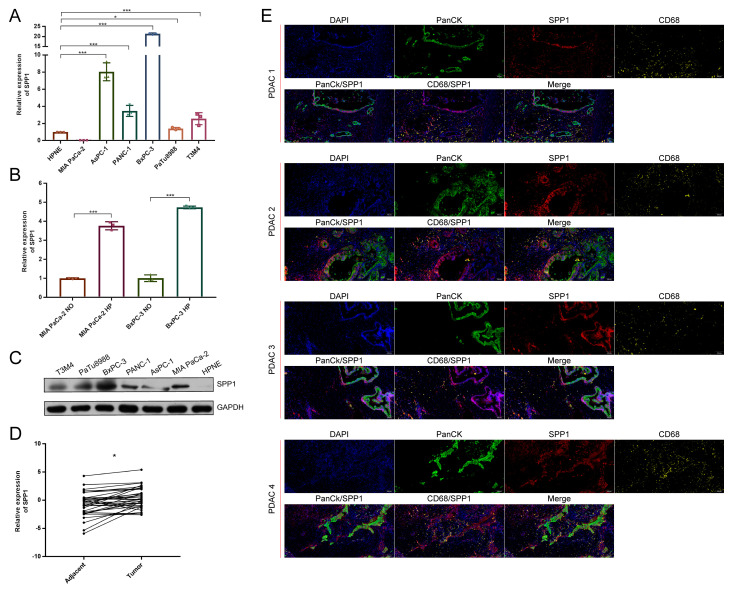
**SPP1 was mainly expressed in ductal and macrophage subpopulations of PDAC.** (**A**) RT-qPCR was performed to show the expression levels of SPP1 in pancreatic cell lines, and HPNE is a normal pancreatic ductal cell line; (**B**) the expression levels of SPP1 under hypoxic and normoxic conditions; (**C**) western blot showing the expression levels of SPP1 among pancreatic cell lines; (**D**) RT-qPCR showing the relative expression of SPP1 between the tumor and adjacent pancreatic tissues; (**E**) multiple color IHC showing the distribution of SPP1 in PDAC. The SPP1, CD68, and PanCK were labeled with different colors. ND, normoxic; HP, hypoxic. *, *** represent *p* < 0.05 and 0.001.

## Data Availability

All scRNA-seq datasets, related codes, and data analysis scripts will be provided upon request to first author Kai Chen (Drchenkai@pku.edu.cn).

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
