# Peer review of "Single Cell RNA-Seq Identifies Immune-Related Prognostic Model and Key Signature-SPP1 in Pancreatic Ductal Adenocarcinoma"

_genes, 2022, doi:10.3390/genes13101760_

Round 1

Reviewer 1 Report

The authors have constructed a prognostic model for PDAC by integrating scRNA-seq and TCGA data. They also showed a correlation between the prognostic model and tumor immune infiltration status. Finally, they showed SPP1's involvement in PDAC. 

The paper was overall easy to follow but needs major improvements in the following areas.

1. Clear description of single-cell data integration and batch-effect removal. They need to deposit the codebase for easy reproducibility.

2. They have shown the involvement of SPP1, but SPP1 was not selected from an unbiased candidate search. They just used prior knowledge of SPP1 to do the analysis. It is necessary to confirm if better or equally good candidates can be discovered from the data.

3. The authors also need to discuss if the composition of immune cells remains comparable across samples. Likewise, how variable are the IRDEGs across the samples?

4. Is it expected that the three studies in Fig 1A would show so few common DEGs? The authors need to explain and rationalize this finding.

5. Figures are not legible. The authors need to use a minimum font size of 8. 

6. Figure S3 D and E -- the authors claimed to show the correlation between SPP1 and other groups of genes; so, it is not clear what the 2-D matrices are showing.

Reviewer 2 Report

In this manuscript, the authors analyzed multiple scRNA-seq datasets for PDAC, and found five immune-related signatures, including SPP1, LINC00683, SNHG10, LINC00237, CASC19 to further develop risk score formula and predict the overall survival of PDAC patients. Finally, they demonstrated that SPP1 was mainly expressed in ductal cells and macrophages in PDAC, and could be a promising biomarker for PDAC diagnosis and prognosis. The findings are interesting and within the scope of the journal. Some concerns are: 

1.      In Figure 6B, what do ND? and HP mean? How does the expression of SPP1 associate with hypoxia?

2.      SPP1 was found to be overexpressed in many tumors, including GBM and Ovarian cancer. The authors should talk about the specificity of this biomarker, and it’s potential roles in mediating tumor progression in the discussion section.   

Author Response

Response to Reviewer 2 Comments:

Point 1: In Figure 6B, what do ND? and HP mean? How does the expression of SPP1 associate with hypoxia?

Response 1: Thank you for this comment. In this study, ND and HP represent the normoxic and hypoxic conditions, respectively. We have added the corresponding information to the figure legend to clarify the meaning of these two abbreviations. Higher expression levels of SPP1 were found in the MIA PaCa-2 and BxPC-3 cells under hypoxic condition than under normoxic condition (Figure 6B). However, the mechanism by which the hypoxic microenvironment regulates SPP1 expression remains to be investigated.

Point 2: SPP1 was found to be overexpressed in many tumors, including GBM and Ovarian cancer. The authors should talk about the specificity of this biomarker, and it’s potential roles in mediating tumor progression in the discussion section.

Response 2: Good point. We have added key information regarding the specificity and potential roles in mediating tumor progression of SPP1 in the Discussion section (Line No. 315-318).

Round 2

Reviewer 2 Report

The authors have satisfactorily responded to all my questions .